# Ultrasound-Based Hepatic Elastography in Non-Alcoholic Fatty Liver Disease: Focus on Patients with Type 2 Diabetes

**DOI:** 10.3390/biomedicines10102375

**Published:** 2022-09-23

**Authors:** Georgiana-Diana Cazac, Cristina-Mihaela Lăcătușu, Cătălina Mihai, Elena-Daniela Grigorescu, Alina Onofriescu, Bogdan-Mircea Mihai

**Affiliations:** 1Unit of Diabetes, Nutrition and Metabolic Diseases, Faculty of Medicine, “Grigore T. Popa” University of Medicine and Pharmacy, 700115 Iasi, Romania; 2Clinical Center of Diabetes, Nutrition and Metabolic Diseases, “Sf. Spiridon” County Clinical Emergency Hospital, 700111 Iasi, Romania; 3Unit of Medical Semiology and Gastroenterology, Faculty of Medicine,, “Grigore T. Popa”, University of Medicine and Pharmacy, 700115 Iasi, Romania; 4Institute of Gastroenterology and Hepatology, “Sf. Spiridon” Emergency Hospital, 700111 Iași, Romania

**Keywords:** type 2 diabetes mellitus, non-alcoholic fatty liver disease, transient hepatic elastography, hepatic steatosis, liver fibrosis

## Abstract

Non-alcoholic fatty liver disease (NAFLD) is the most prevalent liver disease and is the hepatic expression of metabolic syndrome. The development of non-invasive methods for the diagnosis of hepatic steatosis and advanced fibrosis in high-risk patients, especially those with type 2 diabetes mellitus, is highly needed to replace the invasive method of liver biopsy. Elastographic methods can bring significant added value to screening and diagnostic procedures for NAFLD in patients with diabetes, thus contributing to improved NAFLD management. Pharmacological development and forthcoming therapeutic measures that address NAFLD should also be based on new, non-invasive, and reliable tools that assess NAFLD in at-risk patients and be able to properly guide treatment in individuals with both diabetes and NAFLD. This is the first review aiming to outline and discuss recent studies on ultrasound-based hepatic elastography, focusing on NAFLD assessment in patients with diabetes.

## 1. Introduction

Non-alcoholic fatty liver disease (NAFLD) is a chronic liver disorder that is lately becoming a worldwide major public health problem in both adults and children. The high prevalence of metabolic comorbidities such as obesity, dyslipidemia, and diabetes mellitus (DM) that are frequently associated with NAFLD supports the need for increased attention from healthcare providers who should invest in screening and management [1,2].

According to the latest International Diabetes Federation (IDF) Diabetes Atlas data, an estimated 537 million adults worldwide aged 20–79 years are currently living with diabetes, representing 10.5% of all adults in this age group. In 2021, almost 240 million adults had undiagnosed diabetes [3]. Early identification of people with diabetes is key to avoiding or delaying complications and improving quality of life, thus preventing the significant burden on healthcare systems [3].

The estimative global prevalence of NAFLD is 25% of the adult population. More than 50% of persons with type 2 diabetes (T2DM) and 90% of persons with severe obesity have NAFLD [4,5,6]. Approximately 10 to 15% of NAFLD patients from the United States and Europe have advanced fibrosis. Patients with NAFLD have an increased risk of liver-related death, primarily those with histologically proven non-alcoholic steatohepatitis (NASH) [7]. T2DM doubles the risk of hepatocellular carcinoma [5]. The high prevalence of metabolic syndrome (MS) is independently associated with all-cause, liver-specific, and cardiovascular mortality. Other risk factors leading to the increased prevalence of NAFLD are represented by older age and male sex [5]. The large number of NAFLD patients that are potential candidates for progressive liver disease creates challenges in screening and management, mirroring the evolution of cardiovascular disease development on the background of T2DM, obesity, and insulin resistance [8].

NAFLD covers a spectrum of histological conditions, ranging from simple steatosis (non-alcoholic fatty liver, NAFL) to NASH, which can later progress to liver fibrosis, cirrhosis, or hepatocellular carcinoma (HCC). MS is a predictor of hepatic steatosis [9]. Fibrosis is an important prognostic factor in NAFLD. The fibrosis stage is independently associated with increased overall and liver-specific mortality and with higher rates of liver-related complications and liver transplantation; early studies suggest a higher prevalence of NASH and advanced fibrosis stages among patients with T2DM [9,10].

Experts in the field have recently suggested the introduction of a new acronym, MAFLD (metabolic dysfunction-associated fatty liver disease), which reflects the relevant risk factors for this disease, underlining the association between insulin resistance and MS [11,12].

NAFLD and T2DM display a bidirectional relationship wherein these two pathologies have intricate effects on disease progression. On one hand, NAFLD co-existence increases the incidence of T2DM and the risk of developing micro- and macrovascular complications of diabetes [13,14,15]. On the other hand, T2DM is recognized as a risk factor for progressive liver disease, leading to advanced fibrosis, cirrhosis-related complications, and increased liver disease mortality [16]. An extensive meta-analysis of 33 studies carried out between 2000 and 2020, including 501,022 individuals and nearly 28,000 cases of incident diabetes, showed that patients with NAFLD had a higher risk of incident diabetes mellitus than those without NAFLD. This risk increased considerably in individuals with advanced liver fibrosis [17]. The co-existence of NAFLD and T2DM acts synergically to increase the risk for other organ complications, with the highest mortality in NAFLD attributed to a worsened cardiovascular risk profile [18]. It is thus becoming evident that the link between NAFLD and diabetes is more complex than previously believed.

Two steps are needed to diagnose NAFLD. The first step is to assess the existence of hepatic steatosis, either by imaging or biopsy, and then to exclude other causes of liver steatosis such as significant alcohol consumption, long-term use of steatogenic medication, or monogenic hereditary disorders [9]. NAFL’s only feature is fatty liver infiltration that involves more than 5% of hepatocytes, whereas NASH also features inflammation and evidence of hepatocellular injury, with or without fibrosis, in the absence of alcohol consumption (daily intake of less than 20 g in women and 30 g in men) [9,14].

The European Association for the Study of the Liver (EASL), the European Association for the Study of Diabetes (EASD), and the European Association for the Study of Obesity (EASO) recommend ultrasonography as a first-line screening test for NAFL, whilst liver biopsy represents the essential tool for the diagnosis of NASH [14]. However, the limits of invasive biopsy procedures are acknowledged, and no distinct screening information in the existing guidelines refers to the identification of this metabolic liver disease amongst patients with diabetes.

Therefore, this is the first review aiming to identify and analyze the current elastography-based imaging strategy for NAFLD screening and diagnosis, focusing on its applicability in patients with T2DM. This category of patients has become more and more clinically significant, as the increased prevalence of diabetes and obesity have become important public health issues in recent decades. Among them, non-invasive diagnostic tests such as ultrasound-based hepatic elastography are highly needed to replace liver biopsy, to develop a new protocol for screening patients at risk for NAFLD or those with a history of steatosis diagnosed by hepatic imaging/biopsy, and to non-invasively monitor patients with NAFLD and diabetes and their response to treatment.

## 2. Common Approaches in NAFLD Assessment

Measures to limit disease progression must be based on the identification of metabolic risk factors (obesity, dyslipidemia, hypertension, and diabetes), on the assessment of anthropometric indices and laboratory tests (fasting blood glucose, oral glucose tolerance test, HbA1c, complete lipid profile, uric acid, and thyroid markers), on the calculation of related biomarkers (homeostatic model assessment for insulin resistance index, estimated glomerular filtration rate, urinary albumin-creatinine ratio), and on imaging techniques [9,19,20].

Liver-related outcomes are influenced by the advanced stage of fibrosis and not steatosis. Commonly used non-invasive tests (NIT) widely available in clinical practice to estimate fibrosis are represented by the fibrosis-4 (FIB-4) index, NAFLD fibrosis score (NFS), and aspartate aminotransferase (AST) [21,22]. By using such NIT in a large cohort of patients with T2DM, Singh et al. identified a high prevalence of fibrosis [23]. In fact, in line with the European guidelines, these scores should be calculated for every patient with NAFLD [14]. Their use can exclude the presence of advanced fibrosis in 50–67% of patients with diabetes [24]. A new prediction model, diabetes liver fibrosis score (DLFS) was recently developed to help identify patients with diabetes at significant risk for liver-related morbidity: DLFS values over 68.9 maximize specificity (98%) and positive predictive value (86%), while values less than 14.5 maximize sensitivity (95%) and negative predictive value (92%) [25]. Other NIT such as AST to platelet ratio index (APRI) or Hepascore had less accuracy predicting cirrhosis in patients with NAFLD and diabetes [26].

Ultrasonography stands at the forefront as a non-invasive method of screening and diagnosing steatosis in patients with diabetes. The lipid deposits in the liver can be detected by ultrasound when steatosis exceeds 30% of the liver parenchyma, which is visualized as a bright liver echotexture (hyperechoic) blurring the deeper structures [16,27]. An ultrasonography-based study showed that 127 out of 204 patients with T2DM had hepatic steatosis on ultrasound, and 87% of those having consented to a liver biopsy had NAFLD confirmed by histology [28]. The Edinburgh Type 2 Diabetes Study, the first study using ultrasound grading compared with magnetic resonance spectroscopy to determine NAFLD prevalence in a population of patients with T2DM, showed that the disadvantage of ultrasound is its inability to differentiate grade 1 or 2 of steatosis [29]. Therefore, the most accurate method to quantify fat is magnetic resonance imaging (MRI); however, MRI is limited by its high costs and lesser availability and is mainly used in clinical trials [21].

The “gold standard” in diagnosing NAFLD is liver biopsy and histologic examination. However, liver biopsy is limited by its invasive nature, potentially prone to complications such as pain, bleeding, or sampling errors [30]. The difficulties in repeating biopsies to assess changes in hepatic steatosis and fibrosis and in performing them on individuals with high abdominal circumferences require alternative non-invasive assessment tools [9,31].

The main purposes of following-up patients with diabetes are to identify patients with MS and risk of NAFLD, to detect individuals with a worsening prognosis, and to monitor them once the therapeutic strategy is implemented [32,33]. While the evidence for novel and innovative therapy approaches for NAFLD in subjects with T2DM is rising, elastography techniques might have a reliable role in monitoring patients with NAFLD and diabetes and their response to treatment [34]. Liver imaging plays an important role in NAFLD assessment in patients with T2DM because no clinical manifestations exist in the early stages of disease and functional tests may be within normal limits [35].

The recent recommendations by the EASL, EASD, and EASO designate clinical scores, serum biomarkers, and the elastographic evaluation of the liver as NIT accepted for the diagnosis and staging of hepatic fibrosis [21]. The American Diabetes Association (ADA) guidelines also suggest the use of transient elastography (TE) and non-invasive biomarkers for risk stratification [36]. The elastographic method evaluates the presence and severity of liver fibrosis according to the etiology of the liver disease and has already been tested in many liver-related conditions [37,38].

## 3. Ultrasound-Based Hepatic Shear Wave Elastography

Ultrasound-based elastography has found its place among NIT used to screen and assess the severity of NAFLD and is represented by TE, point shear wave elastography (pSWE), and two-dimensional shear wave elastography (2D-SWE) [16]. Within this category, TE is extensively available and can be used as a point-of-care test to estimate liver fibrosis by measuring liver stiffness and hepatic steatosis using controlled attenuation parameter (CAP) measurement [39,40]. The practice guidelines of the Brazilian Society of Hepatology and Brazilian College of Radiology have recently supported the use of elastography, among others, as a tool to assess fibrosis and steatosis in various chronic liver diseases, including NAFLD; due to its accuracy, elastography seems to be a non-invasive and cost-effective alternative to liver biopsy [41].

The screening for undiagnosed non-alcoholic fatty liver disease and non-alcoholic steatohepatitis (SUNN) study suggests that even asymptomatic high-risk individuals should, nevertheless, be screened for NAFLD. Using TE and CAP, Eskridge et al. found that 57% of the study population had steatosis without fibrosis and 16% of them had both steatosis and fibrosis [42]. However, the results of this study likely overestimate the presence of steatosis by using a cut-off value of ≥238 dB/m [43]. Even though specialists have not yet reached a consensus on cut-off values, the EASL guidelines suggest that a CAP value >275 dB/m might be used to diagnose hepatic steatosis [44]. A meta-analysis by Petroff et al. found that the optimal cut-offs when using the XL probe are 297, 317, and 333 dB/m for >S0, >S1, and S2, respectively [43]. Another study showed that the cut-off for S ≥ S2 of 331 dB/m is accurate for the identification of moderate steatosis [39]. Obesity, diabetes, and arterial hypertension proved to be statistically significant risk factors for NAFLD and NASH development [42]. In line with the results of this study, applying such efficient screening strategies to high-risk individuals may help to properly implement therapy and, over time, reduce the burden of NAFLD.

### 3.1. Elastography-Based Imaging Techniques to Assess Hepatic Fibrosis

Ultrasound-based shear wave elastographic methods for the assessment of advanced fibrosis in NAFLD are represented by [37,45]:TE or vibration-controlled transient elastography (VCTE)acoustic radiation force impulse (ARFI) quantification:
◦pSWE (point shear wave elastography)◦2D-SWE (two-dimensional shear wave elastography), or 3D-SWE (three-dimensional shear wave elastography) [46].

#### 3.1.1. Transient Elastography

TE is a non-invasive imagistic technique able to stage liver fibrosis by LSM. The use of TE to estimate liver fibrosis severity was first described by Sandrin et al. in 2003 [47]. Besides being recommended as a clinical diagnostic method in many liver-related conditions such as chronic viral hepatitis, cholestatic diseases, alcoholic liver disease, and autoimmune hepatitis, an accumulating body of evidence supports the use of TE for the diagnosis and staging of liver fibrosis in NAFLD [44,48].

TE is a method non-integrated into standard ultrasound-based systems and performed using the Fibroscan^®^ device (Echosens, Paris, France) that is well correlated with histologically diagnosed liver fibrosis in NAFLD. LSM quantification of liver fibrosis is expressed in kilopascals (kPa) [33,49]. TE can use an M probe for normal-weight patients and an XL probe for patients with obesity [50]. The feasibility of TE using both M and XL probes is 93.5% [51].

According to the Baveno VI consensus, TE has enabled the identification of asymptomatic patients with advanced fibrosis (stage F3–F4) at risk for clinical complications. This consensus has proposed the term “compensated advanced chronic liver disease” (cACLD) as an alternative to chronic liver disease in asymptomatic F3–F4 patients, who are at risk of developing severe portal hypertension. Values between 10 and 15 kPa need confirmation of cACLD, and a value >15 kPa is suggestive of cACLD in the absence of clinical signs [52].

The TE technique has acquired widespread use in clinical studies and daily medical activities, ranging from the screening and diagnosis of hepatic steatosis and fibrosis in patients with suspected NAFLD to the assessment of T2DM prevalence among patients with NAFLD and the follow-up protocols searching for improvements after the initiation of pharmacologic and non-pharmacologic treatment [53].

The Rotterdam Study found the highest probabilities of fibrosis among participants with diabetes and steatosis [54]. It is noteworthy that most studies having investigated the prevalence of NAFLD and its risk factors by the TE tool resorted to non-diabetic cohorts for validating their results, so further studies are needed to stratify the diabetes-associated risk, as optimal cut-offs may be influenced by diabetes mellitus or body mass index (BMI) [55]. On the other hand, the risk of developing diabetes may be influenced by a NAFLD-associated status that evolves over time [56]. A cross-sectional study using TE to evaluate fibrosis among various chronic liver disease populations in a tertiary center in Lebanon appreciated that more than 58% of subjects had NAFLD; also, almost 50% of patients had at least one metabolic risk factor and 20% had T2DM [57].

The performance of NAFLD diagnostic tools among patients with T2DM varies according to the assessment methods. In healthy people, TE measurements of Young’s modulus range from 4.4 to 5.5 kPa [37]. Ahn et al. found a significantly higher LSM in the diabetes group (11.22 ± 10.51 kPa) than in the non-diabetes group (8.07 ± 7.29 kPa), and a higher prevalence of diabetes in patients with NAFLD than in those with chronic viral hepatitis [58]. A cohort study on 283 patients performed by Patel and colleagues revealed 82.5% of them were diagnosed with T2DM and one-fifth with severe obesity; the cut-off values applied for LSM were 8.2 kPa for significant fibrosis, ≥9.5 kPa for advanced fibrosis, and >13 kPa for cirrhosis. In this study, 76.5% of patients with BMI values greater than 40 kg/m^2^ required the use of the XL probe [59]. XL probes are designed for obese patients to improve the measurability of liver stiffness [60]. According to Garg et al., TE using the XL probe has a lower rate of failure than the M probe in patients with obesity, being able to evaluate hepatic steatosis and fibrosis in almost 60% of the obese persons with a BMI ≤ 45 kg/m^2^ [30].

In a cross-sectional trial conducted in Vietnam, assessing diabetic patients by TE, a 73.3% prevalence of NAFLD was found among patients with T2DM. The LSM values in patients with F2 (significant fibrosis), F3 (advanced fibrosis), and F4 (cirrhosis) were ≥7 kPa, ≥8.7 kPa, and 11.5 kPa, respectively. After applying multivariable logistic regression, the investigators found AST and platelets as predictors of advanced fibrosis in patients with T2DM [61]. Therefore, patients with diabetes and increased AST values may be predisposed to increased liver stiffness [62].

The heterogeneity of study results may be influenced by specific BMI and waist circumference cut-off values depending on the country and ethnic origin of patients. The rising rates of obesity, dyslipidemia, hypertension, and MS in people with NAFLD support the need for evaluating MS components in patients with fatty liver, but also for NAFLD screening among patients with metabolic risk factors [63].

Several studies compared the use of TE alone with combined NIT for the detection of fibrosis. The STELLAR study demonstrated that the combined use of two NIT among patients with enhanced liver fibrosis (ELF), NFS, FIB-4, and liver stiffness by TE improved the diagnostic performance by reducing the proportion of patients with advanced fibrosis due to NASH and indeterminate results [64].

Combining clinical scores and serum markers with LSM by TE may facilitate and improve the diagnosis of advanced fibrosis and steatosis [65,66,67]. The Fibroscan-AST (FAST) score combines LSM and CAP measured by TE with aspartate aminotransferase, having already been validated in large global cohorts [68]. Comparison of NIT to accurately identify advanced fibrosis due to NASH subsequently reduces the need for liver biopsy to assess the fibrosis stage [64]. The implementation of such a strategy may be particularly beneficial in high-risk patients such as those with T2DM.

The development of novel therapeutic strategies to improve NAFLD-related outcomes also requires high-value evaluation methods such as LSM. Unlike liver biopsy, this tool is widely available and reproducible, avoids patient reluctance, and can be repeated to monitor the results of pharmacological treatment [69].

As LSM by TE has become the most investigated and embraced method for evaluating NAFLD, forthcoming years will show whether it may be designated as a future “gold standard” among non-invasive assessment tools. Studies focusing on the estimation of liver stiffness with TE in patients with diabetes and NAFLD are described in Table 1.

#### 3.1.2. Point Shear Wave Elastography (pSWE)

This technique, based on ARFI, is integrated into conventional ultrasound systems [101,102]. A significant advantage of pSWE is that it can assess liver fibrosis and evaluate the liver parenchyma on the same examination [102].

Shear wave elastography was also used in measuring liver stiffness in a case series of ten patients with diabetes and dyslipidemia in which the safety and effectiveness of saroglitazar in improving NAFLD, a dual PPAR α/γ agonist approved for diabetes in India, were assessed [103].

When the accuracies of LSM by TE, ARFI, and supersonic shears wave (SSI) for the staging of fibrosis were compared on a cohort of patients with NAFLD using liver biopsy as a reference, ARFI performance was found to be better for severe fibrosis and cirrhosis than for mild to moderate fibrosis. As more than half of the selected population had T2DM, variables such as BMI ≥30 kg/m^2^, waist circumference ≥102 cm, or increased intercostal wall thickness may have interfered with and provided unreliable results when ARFI was used, compared with other imaging techniques [104].

However, mixed results are reported in this area. A meta-analysis assessing the diagnostic performance of pSWE vs. TE for staging liver fibrosis found a higher rate of failure in TE measurements using the M probe, more than in pSWE estimations, and obesity appeared to have a lesser influence on the results (11.3% vs. 0.8%) [105]. Giuffrè et al. screened several subjects with obesity having undergone bariatric surgery and reported that LSM is machine-dependent when taking into consideration the skin-to-liver distance (SLD) effect and not just the BMI [106].

While the Ultrasound Liver Elastography Consensus Statement, of the Society of Radiologists, recommends the “rule of four” (5, 9, 13, and 17 kPa) for liver stiffness cut-off values obtained using pSWE or 2D-SWE in NAFLD, no cut-off values specific to the T2DM population exist [107].

There are limited studies on the diagnostic ability of pSWE in patients with NAFLD and diabetes, and some of them included a limited number of patients with diabetes in the selected population. However, intriguing results reported by Meyer et al. using pSWE revealed a relatively high prevalence of liver fibrosis associated with NAFLD, even in patients with type 1 diabetes mellitus (T1DM), with a rate of 16% vs. 31% in T2DM subjects [108].

Existing studies have predominantly involved populations with obesity (adults and children), of which some underwent bariatric surgery [109,110,111]. Studies focusing on the estimation of liver fibrosis with pSWE in patients with T2DM and NAFLD are described in Table 2.

#### 3.1.3. Two-dimensional Shear Wave Elastography (2D-SWE)

Two-dimensional shear wave elastography, or SSI, is also an ARFI-based technique that seems to be a rapid and reproducible technique that adapts ultrasound imaging to measure liver stiffness [102]. In healthy populations, liver stiffness values found by 2D-SWE range between 4.4 and 4.9 kPa [46].

In the two-center study by Cassinotto et al., the relevant covariates influencing the results of the 2D-SWE method were increased waist circumference, higher BMI values, thicker intercostal wall, and, in some cases, diabetes [104].

MS is associated with high liver stiffness [115]. Moreover, a cross-sectional, one-center Japanese study in people with abdominal obesity (Japanese diagnostic criteria for MS include waist circumference values of ≥85 cm for men and ≥90 cm for women) showed that waist circumference was significantly and independently correlated with liver stiffness measured by 2D-SWE [116]. The advantages of 2D-SWE have become evident in individuals with NAFLD and severe obesity, where its findings showed a higher success rate in comparison with TE, one of the most validated tools available [117]. In patients with clinically severe obesity that were evaluated before and after metabolic surgery by 2D-SWE-based LSM, improved characteristics were seen [118].

A comparison of TE, 2D-SWE, and magnetic resonance elastography (MRE) methods found them to be viable alternatives to liver biopsy for examining hepatic stiffness in 231 NAFLD patients. No differences were found between these techniques in the ability to diagnose the F1–3 stages, but MRE was superior to TE and 2D-SWE in detecting the F4 stage. Patients in this study included more than 60% subjects with diabetes, but no other information about this category was available [119].

In another study, obesity, T2DM, and arterial hypertension were independent predictors of a 2D-SWE value ≥ 8 kPa; patients with T2DM and hypertension exhibited a double risk for a hepatologist referral, while patients with obesity had a threefold risk. Therefore, focusing on patients with these medical conditions may improve NAFLD-related risk stratification [120].

Even though pSWE and 2D-SWE are less available in tertiary hepatology clinics and current evidence in patients with T2DM is limited, they may become a forthcoming routine tool for the screening, diagnostic, and therapeutic follow-up of patients with both NAFLD and diabetes [121].

Studies using the 2D-SWE method to assess liver fibrosis in patients with T2DM and NAFLD are described in Table 3.

### 3.2. Additional Results Obtained by Imaging Methods Complemented with Elastography

CAP uses ultrasound waves to detect and quantify liver fat by measuring the degree of ultrasound attenuation by hepatic steatosis after the initial attenuation in the adipose tissue within the abdominal wall [87,122]. It is an affordable method that can identify and monitor persons at risk for NAFLD. Fibroscan^®^ software added CAP in 2010, so it can assess both fibrosis by LSM and steatosis by CAP at the same time [123]. CAP is derived from the attenuation of the same ultrasound data used to track the shear wave speed [124]. Several measurement algorithms based on the same principle as CAP are available on other ultrasound systems [124] but are less used in studies on patients with NAFLD.

CAP qualifies today as a standardized non-invasive measure of liver steatosis. The clinical use of CAP is limited due to difficulties in establishing optimal cut-offs for every steatosis grade and to the influence of other conditions such as diabetes. It appears that the steatosis prevalence in a specific population, its etiology, BMI values, and the co-existence of diabetes must be taken into consideration when interpreting CAP [125].

Several studies using LSM and CAP support the use of CAP to screen for NAFLD in patients with T2DM [87,89,99]. However, current guidelines do not yet recommend it as a standard routine method to identify NAFLD among asymptomatic, even though high-risk, populations [9]. It is noteworthy that a large number of studies overestimated the grade of hepatic fat by using lower, inappropriate CAP cut-offs, as described in Table 4 [75,76,83,89,90].

In the Vietnamese study previously mentioned in the TE section of this paper, steatosis severity was graded using the following CAP cut-off values: S0 (26.7% steatosis) for CAP ≤ 233 dB/m, S1 (20.5% steatosis) for CAP 234–269 dB/m, S2 (21.8% steatosis) for CAP 270–300 dB/m, and S3 (31% steatosis) for CAP > 301 dB/m [61]. This is another example of a study that used an inappropriate CAP cut-off and overestimated the results.

The simultaneous use of LSM and CAP to assess liver fibrosis and steatosis was brought into the spotlight by their implementation in patients with severe obesity that were candidates for bariatric surgery [126]. Only 60% of subjects were eligible for the use of the XL Fibroscan^®^ probe. The results suggested that TE could estimate significant fibrosis (an LSM cut-off value ≥ 9 kPa) and significant hepatic steatosis (CAP ≥ 305 dB/m). The histological findings of patients who underwent liver biopsies appeared to correlate with LSM and CAP results [127].

As previously mentioned, higher estimates of hepatic tissue stiffness are associated with elevated BMI and waist circumference values. Moreover, Sporea et al. found supplementary associations between waist circumference, BMI, elevated AST, HbA1c, severe steatosis, higher CAP values, and advanced fibrosis [87].

The usefulness of CAP in monitoring therapeutic effects is the objective of several studies [34]. Liraglutide was able to reduce CAP-measured hepatic steatosis in addition to its well-known effects on body weight and plasma glucose control [128]. A study investigating, by TE, the effects of the GLP-1 receptor agonist dulaglutide in patients with T2DM was not able to show a reduction of intrahepatic fat, probably due to the short 12-week period of treatment [129]. Another study with a novel thiazolidinedione (lobeglitazone), using CAP by TE and having a primary endpoint of hepatic fat reduction, found improvements in NAFLD in patients with T2DM [130]: a 65% improvement in steatosis, comparable to the PIVENS trial where 69% of NAFLD patients responded to pioglitazone treatment [130,131]. Shimizu et al. assessed the impact of dapagliflozin, an SGLT-2 inhibitor, on liver steatosis and fibrosis: after 24 weeks of therapy, LSM decreased from 9.49 ± 6.05 kPa to 8.01 ± 5.78 kPa and CAP reduced from 314 ± 61 to 290 ± 73 dB/m in the dapagliflozin group [132].

As most studies did not have CAP available when evaluating liver stiffness, and some other studies used only the CAP software to assess NAFLD, we have chosen to address LSM and CAP separately in this paper. Studies using the CAP method to assess liver steatosis in patients with T2DM and NAFLD are described in Table 4.

## 4. The Place of Elastography-Based Techniques in the Screening Algorithm for NAFLD

As previously mentioned, an accumulating body of evidence supports the systematic use of ultrasound-based elastography for assessing hepatic steatosis and advanced fibrosis in high-risk patients. The appropriate management of patients with NAFLD must rely on accurate identification of fibrosis and steatosis severity [133].

Whether a screening strategy using NIT such as TE for the diagnosis of liver fibrosis is cost-effective is still a matter of debate [134]. Future results of the LiverScreen project, which aims to screen for liver fibrosis in the general population in European countries, will probably answer this question after 2025. If the results of this study help identify groups at high risk for chronic liver disease in the general or high-risk population, particularly in patients with obesity and diabetes, improved prevention of liver complications will perhaps become possible, thus ameliorating the burden on healthcare systems [135].

Until then, alternative non-invasive scores such as NFS or FIB-4 are recommended to rule out advanced fibrosis when TE is unavailable, thus minimizing the costs [136]. Other methods, such as MRE, have better sensitivity and specificity but are limited by cost and availability [44,137]. The selection of NIT for the diagnostic algorithm in low-prevalence populations must be performed by consulting a liver specialist [44].

Preliminary results of an ongoing cross-sectional trial reported that less than 2% of patients with diabetes are screened for liver fibrosis in primary and secondary care. A high proportion of cases in which liver fibrosis was confirmed (80.6%) were identified using serum fibrosis markers associated with TE or liver biopsy [138]. On the other hand, in the cross-sectional study by Park et al., the patients with diabetes benefited from fibrosis screening procedures in primary care, even in the absence of steatosis [139].

Unfortunately, the screening rate is low in this high-risk population, despite the high prevalence of significant liver fibrosis and steatosis among patients with diabetes [138]. Therefore, the systematic implementation of a routine screening algorithm is needed to improve the clinical care of patients with NAFLD and diabetes.

Current practices and guidelines have not yet adopted widespread screening because of the lack of evidence supporting the long-term benefits of screening and a favorable cost-effectiveness ratio [9,139]. This might hinder the identification of population groups at risk for NAFLD. Lomonaco et al. argue that NAFLD represents a public health problem for patients with T2DM by emphasizing the burden of the disease in a population with T2DM unaware of NAFLD that was screened with TE [77]. In line with this, Mansour et al., after demonstrating better identification of NAFLD in this category of patients, advised incorporating FIB-4 and TE as a two-tier assessment approach into the routine annual evaluation of patients with T2DM [79]. However, given the large number of people with diabetes, it is unlikely that clinicians will be able to apply TE to all T2DM patients. Therefore, it is important to identify patients at risk for fatty liver disease progression [99].

Currently, most screening suggestions for people with T2DM include non-invasive scores such as FIB-4 or NFS in association with TE [21,140]; this combination can be used to distinguish between populations at low or high risk for advanced fibrosis (Figure 1).

## 5. Gaps in Knowledge

Different researchers have used various cut-offs to study where elastography is positioned in the NAFLD assessment tree. As specific LSM cut-off values to predict fibrosis stages are not yet acknowledged, the method’s reliability could be impaired. The spotlight falls on the optimal LSM cut-off values used to define severe fibrosis (F3 or F4 stages). F ≥ 3 represents advanced fibrosis, while the F4 fibrosis stage usually suggests cirrhosis [142]. Supplementary difficulties arise from some studies not reporting LSM cut-offs that define different fibrosis stages [57,72,91], while others use different stage appellations that are difficult to correlate with the current standard definitions [81,83,89,116]. The age-adapted cut-offs should also be taken into account to improve the method’s performance. Finally, some studies suggest that the use of lower cut-off values would optimize their negative predictive value.

As yet, patients with diabetes have not been compared directly with non-diabetic control groups in elastography-based investigation protocols. Hence, the same cut-off points were applied to stratify fibrosis and steatosis as in any other NAFLD patient. However, the diagnostic and prognostic accuracy for NAFLD of non-invasive imaging tools is significantly influenced by the presence of diabetes [143]. This category of methods needs, therefore, further investigation and validation in populations with T2DM, among which advanced fibrosis has a significantly rising prevalence.

When focusing on T2DM patients and trying to gather specific information on this NAFLD at-risk group population, researchers need to find the best methods to fill in these substantial knowledge gaps. There are only a few studies that directly targeted the diabetic population, while most research involved a larger population, among which only a subgroup of subjects had T2DM [71,96]. Because patients at risk for NAFLD may frequently have significant fibrosis, which can be overlooked on common ultrasound, especially when normal liver enzymes are associated, supplementary screening approaches should be considered, either in the general population or only in at-risk individuals represented by patients with obesity, T2DM, and MS [143]. It is, therefore, logical to presume that the utility of novel non-invasive assessment tools for NAFLD is of utmost importance, but we must acknowledge for now that their predictive ability is insufficiently demonstrated in diabetes populations [144]. At present, ADA recommends that all patients with prediabetes/T2DM and increased liver enzymes or steatosis on ultrasound should be evaluated for the presence of NAFLD, while the other guidelines have discordant approaches [36]. No guideline clearly states who should be selected for screening, who should do the screening, and which method is best to use.

Among elastography-based methods, most available evidence supports the use of TE, while pSWE and 2D-SWE, which are less available in liver clinics, feature limited data on patients with diabetes. Several studies are currently using the SWE techniques, but the available proof is not yet sufficient to generate recommendations, and the need to continue dedicated research in this at-risk population is still high.

The lack of technical information also narrows the reproducibility of data using LSM and CAP assessment. Many publications do not specify whether one or more operators were involved, if they were trained certified examiners, or if patients respected the examination protocol requiring, at least, a three-hour fast before undergoing elastography. [37,107]. The success rate depends on the operator’s experience, but also other various factors such as age, BMI, visceral fat, or the presence of ascites; the probability of elastography-based methods failing increases in patients who are old, obese, or have ascites [14,37]. The number of exploratory measurements may also differ from one study to the other [37,145].

The method’s applicability to NAFLD can be challenging in patients with obesity because of the high rate of failure in measurement and performance without the use of an XL probe. Some of the existing studies had limitations due to not using the XL probe to perform TE examinations on patients with obesity.

As mentioned before, liver biopsy is unsuited for large-scale applications in the diagnosis of NAFLD [143]. Moreover, the applicability of liver biopsies is limited in patients with T2DM and associated cardiovascular disease that need antiplatelet or anticoagulant therapy. However, this method is still required to confirm the results of non-invasive tools in clinical trials. Beyond designing an optimal, cost-effective algorithm for systematic risk stratification, the management of NAFLD in primary care should, therefore, include procedures to accurately estimate and minimize the need for biopsy.

At present, ultrasound-based elastography devices are not accessible in diabetes care clinics, thus requiring a strong collaboration with hepatologists to implement these new, simpler, non-invasive tools and to limit the use of invasive methods in the future. Among steps already taken in this direction, NIMBLE (non-invasive biomarkers of metabolic liver disease) [146] in the USA and LITMUS (liver investigation: testing marker utility in steatohepatitis) [147] in Europe are two projects looking to integrate non-invasive tools into clinical practice and to offer the scientific community data required to receive uniform acceptance.

## 6. Conclusions

To sum up, the results of recent studies show a high prevalence of NAFLD identified by TE among patients with T2DM. These findings support the need for systematic screening for NAFLD to assess the severity of hepatic steatosis and fibrosis in T2DM patients. Within the group of shear wave elastography-based methods, TE has already acquired a well-deserved place, while ARFI-based techniques have begun to collect scientific evidence supporting their value in NAFLD screening, diagnosis, and monitoring among patients with T2DM.

Priorities of this research field should include the setting of cut-off points adapted to specific situations such as the co-existence of diabetes, assessment of the cost-effectiveness and validation of quality criteria for these imaging methods, the risk stratification based on the fibrosis stage, and evaluation of elastography value in the assessment of therapeutic success. Producing a strategic algorithm to check each of these purposes could help diabetes care specialists and primary care providers. An early diagnosis in high-risk patients and the subsequent implementation of adapted interventions such as lifestyle optimization, lipid-lowering therapy, and antihyperglycemic drugs may have the chance to limit NAFLD and its extrahepatic complications, at least until further effective therapies are developed. Beyond this, supplementary research is needed to completely define all long-term benefits of these ultrasound-based elastography techniques.

## Figures and Tables

**Figure 1 biomedicines-10-02375-f001:**
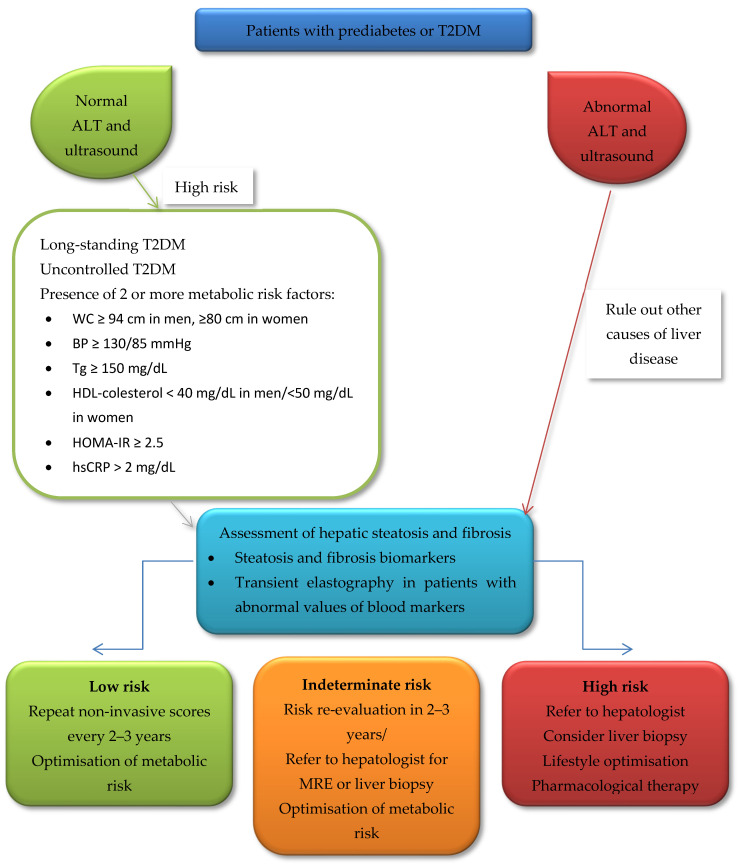
A suggested algorithm to screen patients with T2DM for NAFLD and advanced fibrosis [2,33,63,141]. *Abbreviations:* ALT, alanine aminotransferase; WC, waist circumference; BP, blood pressure; Tg, triglycerides; HOMA-IR, homeostatic model assessment for insulin resistance; hsCRP, high-sensitive C-reactive protein.

**Table 1 biomedicines-10-02375-t001:** Diagnostic performance and comparison of results for different fibrosis stages using LSM by transient elastography in patients with NAFLD and T2DM.

Author, Ref.	Year	Country	No. of Patients	No. of NAFLD Patients	No. of Diabetic Patients	Diabetes Duration (Years)	Mean Age (Years)	Mean BMI (kg/m^2^)	Fibrosis Stage	Cut-Off Level (kPa)
Dai et al. [70]	2022	Taiwan	226	50	226	10 ± 7.8	62.1 ± 10.7	27.3 ± 4.1	F3–4: 50	>7
Trifan et al. [71]	2022	Romania	424	349	424		53.67 ± 11.37	28.07 ± 3.22	F2: 57.14% F3: 11.7% F4: 13.6%	≥8.2 ≥9.7 ≥13.6
Alexopoulos et al. [72]	2021	USA	228 DM	15 5 (TE) unknown NAFLD	228	12.5	58.1	35	F0–1: 40% F2: 20% F3: 40% F4: 0	Unavailable
				Known NAFLD 4 (TE)		15.1	57.9	37.8	F0–1: 25% F2: 25% F3: 50% F4: 0	Unavailable
Cardoso et al. [41]	2021	Brasil	400	173	400	8 (3–15)	64.4	30.4	≥F3: 15%	>9.6
Chhabra et al. [73]	2021	India	200	200	100	–	50.3 ± 11.13	–	F1 F2: 30% F3–F4: 70%	<7 ≥7–8.6 ≥8.7–11.4 ≥11.5
Ciardullo et al. [74]	2021	USA	825	557 steatosis 179 fibrosis	825	9.9 ± 0.75 9.2 ± 2.09 12.9 ± 4.08 10.4 ± 9.23	60.6	31.9 ± 0.47 36.3 ± 1,11 37.5 ± 1.42 38.9 ± 1.45	F0–F1: 76.2% F2: 8.4% F3: 7.7% F4: 7.7%	<8.2 8.2–9.6 9.7–13.5 ≥13.6
Grgurevic et al. [75]	2021	Croatia	454	164	454	–	62.5	30.09	86 45 33	>7.9 ≥9.6 ≥11.5
Gupta et al. [76]	2021	India	250 DM	246 steatosis 205 fibrosis	250	9.6 ± 6.4	51 ± 9	31.4 ± 8	F0: 28.8% F1: 14.8% F2: 18.4% F3: 19.6%	<7 7.1–10 10.1–13 ≥13
Lomonaco et al. [77]	2021	USA	561	70% steatosis 21% fibrosis	561	–	60 ± 11	33.4 ± 6.2	F1: 6.5% F2: 5.6% F3: 6.2% F4: 3%	≥7–8.1 8.2–9.6 9.7–13.5 ≥13.6
Makker et al. [78]	2021	USA	85	-	59	15 ± 9	62 ± 11.7	33.1 ± 8.4	F0–1: 76% F2: 12% F3: 5% F4: 7%	≤7 ≥7.5 ≥10 ≥14
Mansour et al. [79]	2021	United Kingdom	466	58 underwent TE, according to FIB-4	466	–	65.22	33.36	43.1% 20.7% 22.4%	>8 8–15 >15
Sagara et al. [80]	2021	Japan	115	67	115	–	59 ± 13.8	26.6 ± 4.7	F2: 25% F3: 20.5% F4: 13.3%	8–9.6 9.7–12.9 ≥13
Trivedi et al. [81]	2021	USA	437	385	124	–	58.4	33.5	52 100 24	≥7 <10 ≥10
Blank et al. [82]	2020	Germany	204	184	203	13 ± 10.3	64.2 ± 10.7	32.6 ± 7.6	Low 125 Intermediate 10 High 46	<7.9/7.2 M/XL probe 7.9–9.6/7.2–9.3 M/XL probe >9.6/9.3 M/XL probe
Lee CH et al. [83]	2020	China	711	711	711	16.6 ± 9.2	59.4 ± 10.3	28.6 ± 4.5	F0/F1: 40.2% F2: 40.3 % ≥F3: 19.5 %	- - ≥9.6
Lee HW et al. [84]	2020	China	611	Baseline 611	611	–	57.7 ± 10.9	–	63.5% 20%	<10 ≥10
				After 3 years 611					56.5% 4.3%	<10 ≥10
Mantovani et al. [85]	2020	Italy	137	37	137	11	69.9 ± 7	28.5 ± 4.7	F2: 17.5% F3: 10.2%	≥7 ≥8.7
Mikolasevic et al. [86]	2020	Croatia	679	M probe 366 XL probe 313	679	–	65.2 ± 11.6	30.75 ± 5.15	F1: 27.6% F2: 29.5% F3: 29.5% F4: 6.7%	- ≥7 ≥9.6/9.3 M/XL probe ≥11.5/11 M/XL probe
Sawaf et al. [57]	2020	Lebanon	620	362	128	–	47.8 ± 13.4	26.21 ± 4.3	F0–1: 56.6% F2: 9.3% F3: 6.1% F4: 27.9%	Unavailable
Sporea et al. [87]	2020	Romania	776	534	534	10 ± 2	60.8 ± 8.7	32 ± 6	≤F1: 72.6% ≥F2: 7.8% ≥F3: 11.4% F4: 8.2%	- 8.2 9.7 13.6
Tuong et al. [61]	2020	Vietnam	307	18	307	6.5 (3–10)	58.7 ± 11.3	26.3 ± 3.1	F2: 13% F3: 5.9% F4: 3.6%	≥7 ≥8.7 ≥11.5
Arya et al. [88]	2019	India	19,550	6749	13,498	7.52 ± 4.46	50	40% obese 22% overweight 30% normal 8% underweight	F0: 32% F1: 18% F2: 10% F3: 10% F4: 30%	<5.9 6–6.9 7–8.6 8.7–10.2 >10.3
Demir et al. [89]	2019	Turkey	124	31	124	–	53 ± 7	33.2 ± 6.6	≥F3: 16.9% F4: 8%	9.6–11.49.5/9.3–10.9 M/XL probe F4 ≥ 11.5/≥11 M/XL probe
Fernando et al. [90]	2019	Philippines	704	164	285	4.05 ± 3.63	57.27 ± 13.06	27.58 ± 4.25	F0–1: 44.51% F2: 37.8% F3: 5.49% F4: 12.2%	≥5.8 5.9–9.5 9.6–11.5 >11.5
Jaafar et al. [91]	2019	Lebanon	248	248	73	–	53.7 ± 14.6	29.43 ± 7.59	≤F1: 24.66% F2: 17.81% F3: 7% F4: 47.94%	Unavailable
Kumar NA et al. [92]	2019	India	50	47	50	Newly diagnosed	45 ± 4	40% obese	F1: 34% F2: 10% F3: 22% F4: 22% 12%	<5.8 5.8–6.8 6.8–7.8 7.8–11.8 >11.8
Lai et al. [93]	2019	Malaysia	557	403	557	15.8 ± 11.7	60.4 ± 11	29.2 ± 5.2	171 57 37	≥8 M/XL probe ≥9.6/9.3 M/XL probe ≥11.5/11 M/XL probe
Lombardi et al. [94]	2019	Italy	394	350	394	12.3 ± 7.5	65 ± 10	31.4 ± 4.7	83	≥7/6.2 M/XL probe
Wong VW-S et al. [95]	2019	France Hong Kong	496	496	300	–	54 ± 12	30.4 ± 5.4	F1: 112/124 F2: 83/96 F3: 84/91 F4: 59/70	6.8/6.1 M/XL probe 8.8/6.9 M/XL probe 11.8/8.8 M/XL probe 16.3/14.8 M/XL probe
Zhao et al. [96]	2018	China	629 DM	–	629	–	47.07 ± 12.2	26.58 ±4.17	–	F1 > 7.4 F2 > 10.6
Kartikayan et al. [97]	2017	India	60	60	60	7.38 ± 4.2	54.12 ± 11.3	26.6 ± 2.42	F1:16.7% F2:20% F3-F4: 34%	Mean: 7.95
Prasetya et al. [98]	2017	Indonesia	186	84 64 TE	186	<5 y: 38 ≥5 y: 46	<40: 4 ≥40: 80	<25: 25 ≥25: 59	F0-F2: 51 F3-F4: 17	<9.6 ≥9.6
Kwok R et al. [99]	2016	China	1918	334	2119	11.6	61.2	29.3	F3: 17.1%/27.2% F4:11.2%/25	≥9.6–11.4/9.3–10.9 M/XL probe ≥11.5/11 M/XL probe
Sobhonslidsuk et al. [62]	2015	Thailand	197	82	137	–	63.8	27.6	22% 5.93%	≥7 ≥8.7
Ahn et al. [58]	2014	South Korea	979	13	165	–	51.9	25.12 ± 3.11	F0–1: 14% F2/3: 18% F4: 31%	<8 8–19 >19
Casey et al. [69]	2012	Australia	74	26	74	12.2 ± 7.2	61.5 ± 8.6	36.1 ± 5.6	≥F2: 35%	≥7.65
de Lédinghen et al. [100]	2012	France	277	20	277 (132 T2DM)	13	63.2 ± 12.1	27.2 ± 4.3	17	>8.7

*Abbreviations*: NAFLD, non-alcoholic fatty liver disease; BMI, body mass index; kPa, kilopascals; F, fibrosis; S, steatosis; M, medium; L, large; TE, transient elastography; DM, diabetes mellitus; T2DM, type 2 diabetes mellitus; FIB-4, Fibrosis-4 score.

**Table 2 biomedicines-10-02375-t002:** Diagnostic performance and comparison of results for different fibrosis stages using pSWE in patients with NAFLD and T2DM.

Author, Ref.	Year	Country	No. of Patients	No. of NAFLD Patients	No. of Diabetic Patients	Diabetes Duration (Years)	Mean Age (Years)	Mean BMI (kg/m^2^)	Fibrosis Stage	Optimal Cut-Off
Shaji et al. [112]	2022	India	140	30	140	1–5	54.53 ± 12.42	27.37 ± 2.73	21.43%	Unavailable
Meyer et al. [108]	2021	Germany	310	49	T1DM: 93	29	53	25.3	- F2–F4: 8% F3–F4: 5%	1.34 m/s 1.55 m/s 1.8
				88	T2DM: 161	14	65	29.6	- F2–F4: 27% F3–F4: 19%	1.34 m/s 1.55 m/s 1.8 m/s
Demirtas et al. [113]	2020	Turkey	108	54	34	–	54.9 ± 7.7	28 ± 2.2	F1 F2 F3	6.19 ± 1.89 kPa 7.6 ± 1.39 kPa 10.03 ± 4.71 kPa
Roy et al. [103]	2020	India	10	10	10 (T2DM)	7–11	59.3	25.21 ± 3.07	N Mild Moderate Severe Unavailable	1–1.5 m/s 1.5–1.75 m/s 1.75–2.1 m/s >2.1 m/s
Roy et al. [114]	2019	India	36	32	36 (T2DM)	6	52	27.75	N: 11.1% Mild: 27.7% Moderate: 52.7% Severe: 8.3%	1–1.5 m/s 1.5–1.75 m/s 1.75–2.1 m/s >2.1 m/s

*Abbreviations:* NAFLD, non-alcoholic fatty liver disease; T1DM, type 1 diabetes mellitus; T2DM, type 2 diabetes mellitus; BMI, body mass index; F, fibrosis; N, normal.

**Table 3 biomedicines-10-02375-t003:** Diagnostic performance and comparison of results for different fibrosis stages using 2D-SWE in patients with NAFLD and T2DM.

Author, Ref	Year	Country	No. of Patients	No. of NAFLD Patients	No. of Diabetic Patients	Diabetes Duration (Years)	Mean Age (Years)	Mean BMI (kg/m^2^)	Fibrosis Stage	Optimal Cut-Off (kPa)
Miyoshi et al. [116]	2021	Japan	318	-	41	–	63.4	22.7	Unavailable	5.79 ± 1.11
Shaheen et al. [120]	2020	United Kingdom	1958	67 (SWE ≥ 8 kPa)	38	–	61	37.2	91.5% 3.4% 5.1%	<8 ≥8 inconclusive

*Abbreviations:* NAFLD, non-alcoholic fatty liver disease; BMI, body mass index; kPa, kilopascals; SWE, shear wave elastography.

**Table 4 biomedicines-10-02375-t004:** Diagnostic performance and comparison of results for different steatosis degrees using CAP in patients with NAFLD and T2DM.

Author, Ref	Year	Country	No. of Patients	No. of NAFLD Patients	No. of Diabetic Patients	Diabetes Duration (Years)	Mean Age (Years)	Mean BMI (kg/m^2^)	Steatosis Stage	Optimal Cut-Off (dB/m)
Trifan et al. [71]	2022	Romania	424	424	424	–	55.22 ± 10.88	29.12 ± 5.64	S1: 13.1% S2: 8.4% S3: 78.5%	≥274 ≥290 ≥302
Cardoso et al. [41]	2021	Brasil	400	336	400	8 (3–15)	64.4	30.4	41% 22%	>296 >330
Ciardullo et al. [74]	2021	USA	825	557 steatosis 179 fibrosis	825	10.1 ± 0.67 9.8 ± 1.28 15.8 ± 4.16 9.40 ± 1.14	60.6	29.5 ± 0.4 30.3 ± 0.63 34.1 ± 2.72 35.1 ± 0.66	S0: 26.2% S1: 7.2% S2: 8.3% S3: 58.3%	<274 274–289 290–301 ≥302
Grgurevic et al. [75]	2021	Croatia	454	353	454	–	64	30.09	29 22 302	249–268 269–280 >280
Gupta et al. [76]	2021	India	250 DM	246 steatosis 205 fibrosis	250	9.6 ± 6.4	51 ± 9	31.4 ± 8	S1: - S2: - S3: 85.2%	237–259 260–292 >292
Lee CH et al. [83]	2021	China	766	766	766	16.6 ± 9.2	59.4 ± 10.3	28.6 ± 4.5	Mild: 10.2% Moderate: 27.4% Severe: 62.4%	248–267 268–279 ≥280
Lomonaco el al. [77]	2021	USA	561	70% steatosis 21% fibrosis	561	–	60 ± 11	33.4 ± 6.2	S1: 9% S2: 7% S3: 54%	274–289 290–301 ≥302
Makker et al. [78]	2021	USA	85	81	59	15 ± 9	62 ± 11.7	33.1 ± 8.4	S0: 19% S1: 13% S2: 22% S3: 46%	<238 238 259 290
Trivedi et al. [81]	2021	USA	437	213	124	–	58.4	33.5	113 102	≥248 ≥280
Lee HW et al. [84]	2020	China	611	Baseline 611	611	–	57.7 ± 10.9	–	32% 61%	<248 ≥248
				After 3 years 611					12% 52%	<10 ≥10
Mikolasevic et al. [86]	2020	Croatia	679	568	679	7.15 ± 2.33	65.2 ± 11.6	30.75 ± 5.15	83.6%	≥238
Sawaf et al. [57]	2020	Lebanon	620	131	128	–	47.8 ± 13.4	26.21 ± 4.3	S1: 5.2% S2: 7% S3: 45.5%	Unavailable
Sporea et al. [87]	2020	Romania	776	534	534	10 ± 2	60.8 ± 8.7	32 ± 6	S0: 23.9% S1: 8.9% S2: 6.9% S3: 60.3%	- 274 290 302
Tuong et al. [61]	2020	Vietnam	307	225	307	3	56.5 ± 10.5	25.4 ± 2.8	S0: 26.7% S1: 20.5% S2: 21.8% S3: 31%	- 234–269 270–300 ≥301
Demir et al. [89]	2019	Turkey	124	117	124	–	53 ± 7	33.2 ± 6.6	Mild: 0 Moderate: 29 Severe: 88	222–232 233–289 ≥290
Fernando et al. [90]	2019	Philippines	704	164	285	4.05 ± 3.63	57.27 ± 13.06	27.58 ± 4.25	S0: 3.66% S1: 12.8% S2: 39.02% S3: 44.51%	<221 222–232 233–289 ≥290
Jaafar et al. [91]	2019	Lebanon	248	248	73	–	53.7 ± 14.6	29.43 ± 7.59	≤S1 32.3% S2 18.46% S3 27.7% S4 21.54%	Unavailable
Lombardi et al. [94]	2019	Italy	394	238	394	14 ± 8	67 ± 10	29.6 ± 4.2	171 128	≥248 ≥280
Kwok et al. [99]	2016	China	1918	1309	2119	10.7	60.6	26.2	S1: 5.1% S2: 29.6% S3: 38%	222–232 233–289 ≥290
Ahn et al. [58]	2014	South Korea	979	13	165	-	51.9	25.12 ± 3.11	S1: 15% S2: 17% S3: 26%	239–258 259–292 >292

*Abbreviations:* NAFLD, non-alcoholic fatty liver disease; BMI, body mass index; kPa, kilopascals; F, fibrosis; S, steatosis.

## Data Availability

Not applicable.

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
