# Peer review of "Ultrasound-Based Hepatic Elastography in Non-Alcoholic Fatty Liver Disease: Focus on Patients with Type 2 Diabetes"

_biomedicines, 2022, doi:10.3390/biomedicines10102375_

Round 1

Reviewer 1 Report

The aim of this review was “to discuss recent studies on ultrasound-based hepatic elastography used to assess NAFLD in patients with diabetes” and it is claimed that this is the first review article to do so. The content of the article, however, is based also on studies performed in patients with NAFLD, independently on the presence of diabetes. To fulfill the aim, the review should be focused on articles that have assessed the role of the elastography techniques in “NAFLD in patients with diabetes”. Otherwise, the aim and the title of the article must be changed.

There are several mistakes that must be corrected:

The basic information that are given about how shear wave elastography works is completely wrong and the content of table 1 is unacceptable. First of all, the term shear wave elastography includes both transient elastography and the ARFI-based techniques. These latter include pSWE and 2D-SWE. ALL ultrasound shear wave elastography techniques (TE, pSWE and 2-SWE) measure the speed of the shear wave that can be converted to kPa using the Young’s modulus. The statement that pSWE and 2D-SWE are TE methods (line 183) is wrong and unacceptable. The same applies to line 303 (2D-SWE is an ARFI-based technique as well), whereas the statement that 2D-SWE “converts ultrasound imaging to measure liver stiffness” lacks any meaning. Refer to guidelines/consensus for a correct terminology [Refs.#36, 104, Ferraioli G. Ultrasound techniques for the assessment of liver stiffness: a correct terminology. Hepatology 2019;69: 461]. The technique available on the FibroScan device is “transient elastography (TE)” which is also named “vibration-controlled transient elastography (VCTE)”. Be consistent throughout the article, always using one of the two abbreviations and never “Fibroscan” that is not the technique but the device. 

Table 1 must be deleted because it gives wrong information and doesn’t add any value to the review. By the way, VCTE is not “low-priced”: actually, the Fibroscan device costs the same or more than a high-end ultrasound system! Moreover, also pSWE and 2D-SWE are reproducible, and all the techniques share several limitations (some of them are not listed here). By the way, what does it mean “operator-elected”? in pSWE the sample size cannot be changed by the operator.

Baveno VI consensus has highlighted that the spectrum of fibrosis is a continuum in advanced stage and has proposed the term “compensated advanced chronic liver disease” that includes F3 and F4 stages. In this review there isn’t any mention to it.

None information is given about the role of the SWE techniques to evaluate the clinical outcome of patients. This is an area with exciting results.

The title “Elastography-based imaging techniques to assess hepatic steatosis” (line 390) hasn’t any scientific basis. The steatosis quantification tool presented in this section assesses the attenuation of the ultrasound beam and doesn’t have any relationship with elastography. The backscattered signal of the ultrasound beam (this latter being used to assess the speed of the shear waves in the FibroScan device) is used to quantify the attenuation. Elastography and attenuation are unrelated, even though their values are given together. For the above reasons, it is completely wrong to write “CAP by TE”. By the way, it cannot be overlooked that attenuation coefficient measurement algorithms based on the same principle of CAP are available on several ultrasound systems. Check the literature for more information [US attenuation for liver fat quantification: An AIUM-RSNA QIBA pulse-echo quantitative ultrasound initiative. Radiology 2022;302: 495-506; Quantification of liver fat content with ultrasound: a WFUMB position paper. Ultrasound Med Biol 2021;47: 2803-20].

Lastly, recent studies, including a meta-analysis, have shown that the cutoff value for detecting liver steatosis (S>0) with CAP in NAFLD is around 290 dB/m or higher [Assessment of hepatic steatosis by controlled attenuation parameter using the M and XL probes: an individual patient data meta-analysis. Lancet Gastroenterol Hepatol 2021;6: 185-198; Accuracy of FibroScan controlled attenuation parameter and liver stiffness measurement in assessing steatosis and fibrosis in patients with nonalcoholic fatty liver disease. Gastroenterology 2019;156: 1717-30]. In this article there isn’t any mention of these studies. It must be highlighted that the prevalence of NAFLD in the general population estimated using CAP has been LARGELY OVERESTIMATED because the majority of the cited studies used a CAP cutoff of 240 dB/m or even lower (as reported in line 408 and table 5). 

Other comments

Line 33: NAFLD is a not-communicable disease, therefore the use of the verb “to spread” is incorrect.

Lines 39-41: give reference for this statement.

Line 118: AST is not a score

Line 122: sensitivity and specificity??

Line 146-147: If a patient already has NAFLD it is a nonsense to assess the NAFLD risk: please reword.

Line 153: serum biomarkers are not “physical methods”. Please reword.

Line 160: add “shear wave” because other elastography techniques (such as strain elastography) are not validated for this purpose.

Lines 174-177: see general comments: this study has largely overestimated the presence of liver steatosis in the studied population. In fact, in light of the results of recent studies and meta-analysis (see previous comment) a CAP cutoff of 238 dB/cm for the diagnosis of steatosis is inacceptable. It MUST be specified here and thereafter, when reporting studies that used an inappropriate CAP cutoff.

Lines 182-183: this statement is inacceptable (see general comments).

Line 185: FibroScan is a registered name not a “generic” name, and it is the registered name of the device not of the technique.

Line 188: 2D-SWE is an ARFI-based technique as well (see general comments).

Lines 217-218: this sentence is inaccurate. The very first study was published in 2003 in the UMB journal (first author Laurent Sandrin).

Lines 227-229; lines 231-232: these statements are not supported by literature data and MUST be deleted. There isn’t any shear wave elastography technique that is able to accurately detect early fibrosis. The chapter in a book cited here refers to screening for advanced fibrosis and mainly in populations at risk. By the way, keep in mind that ALL shear wave elastography techniques are better at ruling out advanced disease (see guidelines).

Line 233: the phrase “The ultrasound-based technique operating the Fibroscan” is meaningless and must be reworded.

Lines 270-271: this difference is not clinically significant. Please delete.

Line 294: this is a personal thought and must be identified as such.

Lines 303-305: VCTE as well must be performed by trained personnel and it measures the shear wave speed that is then converted to kPa! Check guidelines for the basic information about SWE techniques. This statement must be modified.

Line 326: the rule of four has been proposed by the update to SRU consensus and not by several guidelines. Please correct.

Lines 348-350: the cited study is a two-center study and not a meta-analysis.

Line 391: “CAP by TE” is incorrect. See general comments.

Line 397: highlight here the several limitations of CAP, including uncertainty about the cutoff for steatosis detection.  Other US attenuation algorithms are available (see general comments as well).

Lines 406-409: see general comments about CAP cutoffs.

Lines 417-420: an association doesn’t mean a causation. These sentences must be reworded or deleted.

Lines 440-470: In this section it must be highlighted the fundamental role of blood-based biomarkers to avoid an increase in the health costs without achieving significant benefits. As shown in the figure, blood-based biomarker and elastography are positioned at the same level. Instead, the update to EASL guidelines (ref.#45) has underscored that the selection of noninvasive tests and the design of diagnostic pathways for testing low-prevalence populations for advanced fibrosis should be performed in consultation with a liver specialist. This is important to avoid misinterpretation of the results.

Lines 520-523: are you suggesting that patients must be selected on the basis of their characteristics? It seems quite odd.

Lines 529-531: this sentence has already been repeated several times.

Lines 550-551: the ARFI-based techniques are not a “variant” of TE! Reword.

Line 557: specify why it’s so important to precisely define each stage of liver fibrosis (see general comments).

Reviewer 2 Report

Authors should add that TE is a reliable technique, mainly because it permits early diagnosing the evolving forms of NAFLD, i.e., fibrosis and NASH, which can be likely cured due to the fact that there are many drugs on the pipeline that are good candidates, as evident in various recent papers, for example... Insights into the molecular targets and emerging pharmacotherapeutic interventions for nonalcoholic fatty liver disease. Metabolism. 2022 Jan;126:154925. doi: 10.1016/j.metabol.2021.154925. Epub 2021 Nov 2. PMID: 34740573.

If it is possible, a cost benefit analysis should be presented (beyond those already presented among the various types of TE) in comparison with basic US, and other imaging tools, such as  CT volumetry, as evident in...Fibrosis in nonalcoholic fatty liver disease: Noninvasive assessment using computed tomography volumetry. World J Gastroenterol. 2016 Oct 28;22(40):8949-8955. doi: 10.3748/wjg.v22.i40.8949. PMID: 27833386; PMCID: PMC5083800.......and RMN elastography, which bears a very good sensitivity and specificity, i.e., 100% and 96. 5%, respectively,  as evident in....Non-invasive detection of liver fibrosis: MR imaging features vs. MR elastography. Abdom Imaging. 2015 Apr;40(4):766-75. doi: 10.1007/s00261-015-0347-6. PMID: 25805619; PMCID: PMC4739358.

VCTE, being widely available and characterised by low price is a little bit exaggerated...confronted with ...I would say..... some other indices, such as Fib-4 and APRI or mainly BARD (considering the presence of T2DM as factor to calculate it), as evident in....Development and validation of a simple NAFLD clinical scoring system for identifying patients without advanced disease. Gut. 2008 Oct;57(10):1441-7. doi: 10.1136/gut.2007.146019. Epub 2008 Apr 4. PMID: 18390575.

Reviewer 3 Report

I think Ultra sound is a technique that is non-invasive and can be rolled out to diagnose many patients. Therefore, a meta-analysis such as the reviewed paper, has its place in Literature. I think the Manuscript is well written and covers the field brilliantly.

Compared to biopsy, the ultra sound method is less invasive for the patient.

I recommend publication of this paper

Round 2

Reviewer 1 Report

Line 183: I’d suggest to reformulate this sentence deleting “controversial” and adding “likely” as follows:  the results of this study likely overestimate the presence of steatosis by using a cut-off value of ≥ 238 dB/m. Moreover, for readers not aware of this topic, it must be explained here why this cutoff overestimates steatosis. I’d suggest to move here the sentences from line 427 to line 433.

Line 192: abbreviations must be spelled out at their first mention.

Line 342: please substitute “belonging to” with “of”.
